# Nutritional Status at Diagnosis as Predictor of Survival from Childhood Cancer: A Review of the Literature

**DOI:** 10.3390/diagnostics12102357

**Published:** 2022-09-28

**Authors:** Maria A. Karalexi, Georgios Markozannes, Christos F. Tagkas, Andreas Katsimpris, Xanthippi Tseretopoulou, Konstantinos K. Tsilidis, Logan G. Spector, Joachim Schüz, Tania Siahanidou, Eleni Th. Petridou, Evangelia E. Ntzani

**Affiliations:** 1Department of Hygiene and Epidemiology, School of Medicine, University of Ioannina, 45110 Ioannina, Greece; 2Hellenic Society for Social Pediatrics and Health Promotion, 11527 Athens, Greece; 3Department of Epidemiology and Biostatistics, School of Public Health, Imperial College London, London SW7 2BX, UK; 4Department of Pediatric Endocrinology, Addenbrooke’s Hospital, Cambridge CB2 0QQ, UK; 5Department of Pediatrics, Division of Epidemiology & Clinical Research, University of Minnesota, Minneapolis, MN 55455, USA; 6Section of Environment and Radiation, International Agency for Research on Cancer (IARC), 69372 Lyon, France; 7First Department of Pediatrics, National and Kapodistrian University of Athens, 11527 Athens, Greece; 8Department of Hygiene, Epidemiology, and Medical Statistics, School of Medicine, National and Kapodistrian University of Athens, 11527 Athens, Greece; 9Center for Evidence Synthesis in Health, Brown University School of Public Health, Providence, RI 02903, USA

**Keywords:** undernutrition, obesity, nutritional status, childhood cancer, overall survival, event-free survival, relapse, treatment-related toxicity, review

## Abstract

Few studies so far have examined the impact of nutritional status on the survival of children with cancer, with the majority of them focusing on hematological malignancies. We summarized published evidence reporting the association of nutritional status at diagnosis with overall survival (OS), event-free survival (EFS), relapse, and treatment-related toxicity (TRT) in children with cancer. Published studies on children with leukemia, lymphoma, and other solid tumors have shown that both under-nourished and over-nourished children at cancer diagnosis had worse OS and EFS. Particularly, the risk of death and relapse increased by 30–50% among children with leukemia with increased body mass index at diagnosis. Likewise, the risk of TRT was higher among malnourished children with osteosarcoma and Ewing sarcoma. Nutritional status seems to play a crucial role in clinical outcomes of children with cancer, thus providing a significant modifiable prognostic tool in childhood cancer management. Future studies with adequate power and longitudinal design are needed to further evaluate the association of nutritional status with childhood cancer outcomes using a more standardized definition to measure nutritional status in this population. The use of new technologies is expected to shed further light on this understudied area and give room to person-targeted intervention strategies.

## 1. Introduction

Cancer is the leading cause of death in children and adolescents worldwide [1]. Every year, just over 150,000 children are diagnosed with cancer [2]. Since a large proportion of childhood cancers in low- and middle-income countries are never diagnosed, the more realistic annual number is estimated to be at least twice as high, i.e., above 360,000 children [3]. Given these caveats, the age-standardized incidence (ASR) of the disease, estimated at 140.6 per million person-years in children aged 0–14 years, is increasing, with leukemia being the most common cancer site (ASR: 46.4), followed by central nervous system (CNS) tumors (ASR: 28.2) and lymphomas (ASR: 15.2) [4]. Over the past decades, advances in childhood cancer care, including novel imaging techniques for diagnostics and risk stratification, as well as multimodal novel therapies and supportive healthcare, have led to impressive increases in five-year survival, which exceeds 80% for all cancer types and 90% for acute lymphoblastic leukemia (ALL) in many European and North American countries nowadays [5]. However, severe or life-threatening long-term consequences may occur in up to 80% of childhood cancer survivors [5,6]. Indeed, a recent report showed that mortality among cancer patients is higher than that in the general population, mainly attributed to increased cardiotoxicity and risk of second neoplasms [7]. Poor nutritional status, defined as undernutrition (body mass index (BMI) < 5th percentile) or overnutrition (BMI ≥ 85th percentile) by the World Health Organization (WHO), seems to be a poor prognostic indicator linked to treatment-related toxicities in adults with solid tumors [8]. Recent research suggests that nutritional factors may also adversely affect outcomes in children and adolescents treated for cancer [9]. However, few studies have examined the impact of nutritional status on the survival of children with cancer, with the majority of them focusing on hematological malignancies [10,11].

Undernutrition is commonly reported in children treated for cancer, with a prevalence rate estimated as high as 50% in some populations [12]. Multimodal dose-intense antineoplastic therapies, surgery, and radiotherapy may cause serious complications that synergistically enhance poor nutritional status [13]. Treating undernutrition in children with cancer will assist them in tolerating traditional and novel therapies and thus plays a crucial role in improving outcomes and quality of life [13,14,15].

Overnutrition has long been associated with treatment-related complications. Such examples include the increased risk for thrombo-hemorrhagic fatal events in children with acute promyelocytic leukemia, the increased risk for infections (i.e., urinary tract infections and infections of central lines) in patients with ALL, as well as increased nephrotoxicity and post-operative complications in patients with osteosarcoma [15,16,17,18].

We aimed to review published literature and summarize evidence reporting the association of nutritional status at diagnosis with overall survival (OS), event-free survival (EFS), relapse, and treatment-related toxicity (TRT) in children treated for different types of cancer.

## 2. Methods

An independent and blinded literature search of the Medline database (via PubMed) was conducted from inception up to October 18 2021 using an algorithm based on relative key terms, such as “nutritional”, “undernutrition”, “obesity”, “BMI”, “prognostic”, “survival”, “relapse”, “toxicity”, “complications”, “childhood malignancies”, and “childhood cancer”. The reference lists of identified studies were searched for additional eligible articles potentially missed through the initial literature search in a procedure called “snowball” procedure. Eligible were studies that examined the association of nutritional status at diagnosis with clinical outcomes, such as OS, EFS, relapse, histological response, and TRT in children (0–14 years) with cancer. We considered both systematic reviews and meta-analyses, as well as primary publications eligible for inclusion. Other types of publications, such as letters to the editor, commentaries, and editorials, were excluded. Case reports, case series, in vitro, or animal studies were also excluded. Nutritional status was defined by each study based on body mass index (BMI), weight change, or other body composition indices that defined under- or over-nutrition. We included studies reporting on any type of pediatric cancer.

The identified articles were independently screened by two reviewers to identify those that met the pre-determined inclusion criteria. Disagreements in the selection of studies or snowball procedure were resolved by team consensus. In articles with overlapping populations, the most recent or most complete publication was considered eligible. We extracted data based on a pre-defined form, including the name of the first author, publication year, study type, childhood population characteristics (including the studied cancer sites), the definition of the nutritional status based on the study, outcomes of interest (OS, EFS, relapse, TRT) along with the accompanying effect estimates (relative risk, hazard ratio, etc.) and their corresponding 95% confidence intervals (CI) and/or *p*-values.

Due to the large heterogeneity in exposure and outcome definitions across the identified studies, no quantitative synthesis of the results was feasible. Thus, a narrative presentation of the eligible studies by cancer type was performed (Table 1).

## 3. Results

The search strategy identified 4927 records, and after title, abstract, and full-text screening we included 22 publications in the present review (Figure 1). We identified 4 systematic reviews and meta-analyses and 18 primary studies. The most commonly studied cancer site was acute leukemia, reported in six publications (27%), followed by all cancer sites as a combined variable, which was reported in four primary studies (18%). The most commonly studied outcome was OS (n = 17 studies; 77%), followed by EFS (n = 10 studies; 45%), while treatment-related complications were studied as outcome in seven publications (32%).

### 3.1. Hematological Malignancies

A previous meta-analysis of 11 articles found poorer EFS in children with ALL who had higher BMI (≥85th percentile; relative risk [RR]: 1.35; 95% CI: 1.20, 1.51; n = 6 studies; I^2^: 34%, *p* = 0.18) compared to those with lower BMI at diagnosis (Table 1). In addition, children with higher BMI at diagnosis had significantly higher mortality (RR: 1.31; 95% CI: 1.09, 1.58; n = 4 studies; I^2^: 49%, *p* = 0.12) [19]. Between-study heterogeneity was non-significant in all meta-analyses, whereas no evidence of publication bias was shown. Children with acute myeloid leukemia (AML) and higher BMI at diagnosis were also associated with poorer EFS (RR: 1.36; 95% CI: 1.16, 1.60; I^2^: 0%, *p* = 0.99) and OS (RR: 1.56; 95% CI: 1.32, 1.86; I^2^: 0%, *p* = 0.66) compared to children with lower BMI [19]. Similarly, two other meta-analyses showed significant associations between obesity and childhood leukemia survival outcomes. In particular, a meta-analysis including 11 studies found a statistically significant association between obesity, as defined by high BMI at diagnosis, and poorer OS (hazard ratio [HR]: 1.30, 95% CI: 1.16, 1.46; n = 7 studies; I^2^: 71%; *p* = 0.002); however, statistically significant between-study heterogeneity was noted in this meta-analysis. High BMI at diagnosis was also associated with poorer EFS (HR: 1.46, 95% CI: 1.29, 1.64; n = 7 studies; I^2^: 39%, *p* = 0.13) among children with acute leukemia [20]. A more recent meta-analysis (2018) showed an increased mortality rate (HR: 1.79, 95% CI: 1.03, 3.10; n = 3 studies; I^2^: 0%, *p* = 0.37), as well as increased risk for relapse (HR: 1.28, 95% CI: 1.04, 1.57; n = 2 studies; I^2^: 0%, *p* = 0.59) for children older than 10 years with ALL. However, these meta-analyses included a limited number of studies (n = 2–3) [21]. A statistically significant association between obesity and increased mortality for AML was also shown (HR: 1.64, 95% CI: 1.32, 2.04; n = 3 studies; I^2^: 0%, *p* = 0.73), whereas no effect sizes were reported for AML relapse [21]. A recent study from Brazil assessed the impact of nutritional status on outcomes of children (n = 148) undergoing allogeneic hematopoietic stem cell transplantation (HSCT). Severe malnutrition was linked to a statistically significantly increased risk of acute graft versus host disease (HR: 1.68, 95% CI: 1.02, 2.74), as well as increased mortality (HR: 3.63, 95% CI: 1.76, 7.46), poorer progression-free survival (HR: 2.12, 95% CI: 1.25, 3.60), and poorer OS (HR: 3.27, 95% CI: 1.90, 5.64) [22]. A meta-analysis including 24 studies (18 on adolescents and 6 on children) examined the association between BMI and clinical outcomes after HSCT in patients with hematological malignancies. Compared to normal, lower BMI before or during transplantation was significantly linked to poorer OS (RR_pre-HSCT stage_: 1.17, 95% CI: 1.08, 1.27; n = 11 studies; I^2^: 0%, *p* = 0.67/RR_HSCT stage_: 1.34, 95% CI: 1.01, 1.78; n = 5 studies; I^2^: 60%, *p* = 0.04), as well as to poorer EFS (RR_pre-HSCT stage_: 1.29, 95% CI: 0.96, 1.72; n = 5 studies; I^2^: 18%, *p* = 0.30/RR_HSCT stage_: RR: 1.53, 95% CI: 1.09, 2.06; n = 2 studies; I^2^: 31%, *p* = 0.23). Again, the small number of included studies in some meta-analyses should be acknowledged. By contrast, there was no impact of high BMI on clinical outcomes among these patients at any transplantation stage [23].

A recent study in Pakistan showed that OS in children with Hodgkin lymphoma significantly decreased in cases of moderate (79%) and severe malnutrition (75%) compared to no malnutrition (96%; *p* = 0.006) [24]. By contrast, a 2021 study including 191 children with leukemia and lymphoma showed no impact of malnutrition on clinical outcomes [25].

The small sample sizes and the retrospective design of the identified studies should be acknowledged when interpreting these results. However, overall evidence from published studies suggests that poor nutritional status, especially over nutrition at diagnosis, may be a significant predictor of poor outcomes among children with leukemia, particularly ALL. Further research is needed to investigate the association with other leukemia subtypes, especially the rarer AML subtype, as well as the potential association with lymphomas, where evidence remains inconclusive.

### 3.2. Ewing Sarcoma

A USA cohort study including data from the University of California, San Francisco (UCSF) and Stanford University Medical Centers (n = 142 patients with Ewing sarcoma) showed non-statistically significant correlations between BMI at diagnosis and TRT (*p* = 0.43), specifically grade 3 and grade 4 non-hematologic toxicities during follow-up [26] (Table 1). With regards to clinical outcomes, in a study from Israel, abnormal BMI, defined as high or low BMI combined into one category, was statistically significantly associated with poor histologic response, namely tumor necrosis < 90% (odds ratio [OR]: 4.33, 95% CI: 1.12, 19.14), as well as worse OS (HR: 2.76, 95% CI: 1.19, 9.99) [27], whereas no correlations were found with EFS in this population (n = 50 patients). Lastly, regarding TRT, a study from Canada (n = 71 patients) reported significant correlations between low BMI at diagnosis and cardiotoxicity among children receiving anthracycline chemotherapy for Ewing sarcoma (*p* = 0.03), though this association was non-significant in multivariate logistic regression models (*p* = 0.35) after adjusting for potential covariates [28]. Given the limited sample sizes and the retrospective design of the identified studies, further research is needed to allow firm conclusions to be drawn regarding the association of nutritional status with Ewing sarcoma outcomes.

### 3.3. Osteosarcoma

Two longitudinal cohort studies derived from the Children’s Oncology Group (COG) examined the impact of nutritional status on TRT and clinical outcomes in patients with osteosarcoma [29,30] (Table 1). Regarding TRT, the first study (n = 498) showed that high BMI was significantly associated with an increased risk of arterial thrombosis in the post-operative period (OR: 9.40, *p* = 0.03) [29]. The second study showed that children with high BMI had increased odds of developing grade III–IV nephrotoxicity (OR = 2.70, 95% CI: 1.20, 6.40). Regarding clinical outcomes, children in the high BMI group had a significantly poorer 5-year OS of 70% compared to children with normal BMI (OS: 80%; HR: 1.60, 95% CI: 1.14, 2.24). There was also a trend towards worse EFS at 3 years from diagnosis in children with high BMI compared to those with normal BMI (66% versus 75%; HR: 1.30, 95% CI: 0.90, 1.80), whereas there was no impact of low BMI on OS and EFS among these patients [30].

### 3.4. Rhabdomyosarcoma

Two studies based on prospective data obtained from the COG assessed the association between nutritional status and rhabdomyosarcoma outcomes [31,32] (Table 1). In the first study, nutritional status was not a prognostic indicator for infections or survival. However, patients with more than 10% weight loss at 24 weeks of follow-up had a significantly increased number of hospitalization days (OR: 1.24, 95% CI: 1.00, 1.54) [32] and a trend towards a higher risk of grade 3 and grade 4 toxicity (OR: 1.16, 95% CI: 0.99, 1.35) at 42 weeks of follow-up. Children with low BMI (<10th percentile) did not have significantly worse OS compared to those with normal BMI at baseline (HR: 1.70, 95% CI: 0.98, 2.96; *p* = 0.0596). The second USA study on 570 children with intermediate-risk rhabdomyosarcoma examined the impact of tumor volume and patient weight on EFS based on a partitioning algorithm and accounting for age and greatest tumor dimension. The results of the algorithm showed that tumor volume (≥20 cm^3^), histology, and patient weight (≥50 kg) were statistically significantly associated with twofold worse EFS (HR: 2.12, 95% CI: 1.12, 4.02) [31].

### 3.5. Other Cancer Sites

Two studies reported non-statistically significant associations between nutritional status and OS in children with neuroblastoma and Wilms tumor [33,34] (Table 1). A study from Turkey evaluated the impact of nutritional status on the survival of children with all types of cancer. Though malnutrition was a significant complication in this population, with a prevalence of 30% at diagnosis increased to 38% three months later, there was no impact of malnutrition on survival outcomes [35]. Likewise, a study on 139 children with Ewing sarcoma and osteosarcoma reported high proportions of malnutrition 2 years after treatment initiation (43% of osteosarcoma and 25% of Ewing sarcoma) [36]; again, malnutrition had non-significant effects on survival outcomes. A more recent study assessed the prognostic effect of sarcopenia, defined by the BMI-z score, the prognostic nutritional index (PNI), and the total psoas muscle area (tPMA), on the survival of children with bone and soft tissue sarcomas [37]. This study showed that the decrease in PNI (*p* = 0.03) and tPMA of more than 25% (*p* = 0.04) were associated with worse one-year OS; yet, more research is needed to evaluate the prognostic significance of tPMA in children with sarcomas. A study on pediatric cancer patients in Hungary showed that undernutrition at diagnosis, defined by the BMI Z-score and the ideal body weight percent (IBW%), was associated with worse 5-year OS only in patients with solid tumors [38]. Lastly, a study from the Netherlands, including 269 children with all types of cancer, found a significant impact of malnutrition on OS (HR: 3.63, 95% CI: 1.52–8.70), whereas increased weight loss (>5%) was also linked to increased risk of febrile neutropenia and subsequent bacteremia during the first year following diagnosis (OR: 3.05, 95% CI: 1.27, 7.30) [39]. Similar findings have been reported by a recent study in Italy which showed that weight loss of more than 5% at 3 months following diagnosis was associated with poorer OS (HR: 2.75, 95% CI: 1.12, 6.79) and a higher risk of infections requiring hospitalization (HR: 7.72, 95% CI: 2.27, 26.2) [40].

## 4. Discussion

The present review identified 22 published studies reporting associations between nutritional status and survival outcomes in children diagnosed with cancer. Despite the large heterogeneity in studied exposures and outcomes, the present findings suggest that malnutrition is a significant predictor of outcomes in childhood cancer. In particular, published studies on children with acute leukemia, lymphoma, and other solid tumors have shown that both under-nourished and over-nourished children at cancer diagnosis had worse OS and EFS. Of note is that the risk of death and relapse increased by 30–50% among children with leukemia with increased BMI at diagnosis. Likewise, the risk of TRT was higher among malnourished children with osteosarcoma and Ewing sarcoma.

Over the last decades, epidemiological and clinical research has aimed to identify prognostic indicators of childhood cancer with the ultimate goal of optimal risk stratification and targeted therapeutic interventions. Several sociodemographic and clinical factors have been reported to affect the prognosis of childhood cancer, such as age, socioeconomic status, disease subtype, cytogenetic and molecular markers, laboratory markers, predisposing syndromes, tumor size, presence of metastases, etc. [41,42]. However, very few environmental factors have been investigated as potential predictors of outcomes in children with cancer [43,44]. Among these factors, nutritional status is a modifiable marker that has long been postulated to affect survival in both adult and pediatric malignancies [45]. Potential underlying mechanisms that may underlie the association between nutritional status and health outcomes include its impact on body composition makeup, modification of tumor microenvironment, and potential alteration of chemotherapy pharmacokinetics [46]. In particular, changes in lean tissue and fat mass can alter the distribution of chemotherapeutic agents, modify their metabolism, and subsequently affect their clearance, especially the clearance of hydrophilic and/or lipophilic drugs from systemic circulation.

### 4.1. Undernutrition and Cancer Outcomes

Undernutrition has been associated with treatment-related adverse effects and survival among adult patients with cancer [46]. Several underlying biological mechanisms have been proposed to explain this association. Weight loss and decreased muscle and fat mass can be induced by cancer progression, and they both result in a negative balance between synthetic and degradative protein pathways. Moreover, weight and muscle loss are responsible for the activation of inflammatory response and apoptosis of myocytes, as well as for the decreased muscle regeneration capacity [47]. In addition, muscle loss may decrease the excretion of anabolic hormones, such as testosterone and insulin-like growth factor (IGF)-1 [48]. Such phenomena may alter drug pharmacokinetics and thus contribute to the increased risk of dose-response toxicity, which is often observed in cancer patients with low muscle and lean tissue mass [48]. Though evidence in children with cancer is limited, previous studies have shown significant skeletal muscle mass wasting with a concurrent fat mass increase in children with ALL following treatment initiation [49]. Muscle wasting has been related to reductions in chemotherapeutic doses, delays of treatment, and/or premature treatment cessation [50]. Similar findings have been reported by studies on children with several types of solid cancer, concluding that body composition markers seem to be crucial prognostic indicators among pediatric oncology patients [49,51].

### 4.2. Overnutrition and Cancer Outcomes

With regard to overnutrition at diagnosis and survival from childhood cancer, several potential mechanisms have been proposed. There is clinical evidence suggesting that obese patients may have altered pharmacokinetics in the metabolism of chemotherapeutic agents, although prospective trials have not confirmed such variations [52,53]. Overnutrition may also affect both the tumor microenvironment and the microenvironment of the host [54]. In particular, the adipose tissue microenvironment physiology in obese patients is characterized by increased inflammation, as well as disruptions in vascularity and fibrosis which may contribute to tumor progression [54]. Moreover, obesity is associated with hyperinsulinemia and peripheral insulin resistance, which result in reduced growth hormone (GH) secretion and subsequent activation of the IGF-I system as a response to the decreased GH [55]. The association of the IGF-II axis with obesity remains less clear. Low IGF-II levels have been reported in obese children, especially in those with insulin resistance and increased inflammatory markers, such as IL-6 and TNF-a [56]. The overexpression of the IGF-I and -II axis in obese individuals seems to also play a crucial pathogenetic role in the first (initiating) “molecular hit” that contributes not only to leukemia development but also to the disease clinical course [57,58]. Indeed, numerous epidemiologic studies have supported this hypothesis, showing that both children of high birthweight and patients with diabetes have high expression of the IGF-I and II systems and are at increased risk of developing leukemia [18,59]. In addition, these metabolic and chronic inflammatory changes in obese cancer patients may result in the activation of multiple molecular cancer pathways, which subsequently increase the risk of disease resistance and progression [60].

### 4.3. Nutritional Interventions

Apart from the direct effect of cancer itself on nutritional status, several factors can also affect both appetite and food intake among cancer patients. Such factors include some common complications of cancer treatment, such as nausea, vomiting, mucositis (oral, esophageal, and bowel), diarrhea, and/or constipation [61]. Thus, targeted interventions to improve the nutritional status of children with cancer remain challenging. Dietary support and physical exercise recommendations both aim to maintain and improve the normal physical growth of patients before and during treatment [62]. A recent study evaluated the nutritional status following the implementation of a nutritional algorithm in children with cancer from low- and middle-income countries [63]. This study showed an increase in the mid-upper-arm circumference in children enrolled in the algorithm protocol compared to those receiving the standard nutritional care provided by their institution (*p* = 0.02); however, a non-significant difference in weight change was noted between the two comparison groups (*p* = 0.15) [63]. Two other trials assessed the prognostic effect of ω-3 fatty acids and black seed oil on methotrexate-induced toxicity in children with ALL showing a decrease in the risk of hepatotoxicity during the maintenance treatment phase [64,65]. Other interventions that have been explored include the use of glutamine in order to sustain the integrity of the mucosal cell and gut barrier, which is often damaged in chemotherapy-related mucositis [66]. Though evidence is scarce in pediatric patients, one study has shown a significant decrease in the use of parenteral feeding in children with cancer after the exogeneous administration of glutamine without any adverse effects; yet the incidence of mucositis was not reduced in the intervention group [66]. Novel alternative avenues to be explored include the potential beneficial effect of antioxidants or a ketogenic diet on clinical outcomes of children with cancer; however, further research is warranted in this understudied area [67].

Overall, treatment of under- and over-nutrition in cancer patients is not easy on clinical grounds. Interventions should be proactive, aiming to prevent the nutritional depletion before it becomes clinically apparent [68]. Parents and guardians of patients are important components in supporting the nutritional status of children with cancer throughout therapy [69]. Educational and dietician’s support is also crucial in optimizing the health care support of little patients with cancer [70]. Currently, several approaches are being implemented targeting the family-based nutrition and cooking education of parents and guardians of children with cancer [71,72,73]. Among such interventions, web-based dietary and cooking intervention approaches have gained great attention among parents and young patients [74,75,76]. Preliminary results have shown that early interventions resulted in better nutrition practices, i.e., lower sodium consumption [73]. However, obstacles to the participation of families have been noted by several studies, mainly due to treatment-related complications affecting the child’s health, requirement of parents’ presence at the hospital, as well as time, financial and other logistics restraints [71,72]. Future research is thus needed to address these barriers and encourage nutrition and cooking education in a family-based manner.

### 4.4. Methodological Considerations

The limited power of published studies is an important methodological issue when evaluating the potential association between malnutrition and survival of pediatric cancer. Even in the case of meta-analysis, the limited number of included studies (less than 10 studies) in the majority of meta-analyses performed should be acknowledged. Moreover, most studies focused on the most frequent cancer site, acute leukemia, whereas evidence on the remaining cancer sites is scarce, thus not allowing firm conclusions to be drawn. In addition, the large heterogeneity in exposure and outcome definition should be acknowledged. The majority of identified studies considered nutritional status by measuring BMI. Other nutritional indices were weight change, BMI z-score, prognostic nutritional index (PNI), and total psoas muscle area (tPMA). Of note is that recent research argues for the use of BMI-related measures to evaluate the lean mass in cancer patients. By contrast, recent studies have proposed other mainly imaging-based measures, such as cross-sectional computed tomography, which seem to assess sarcopenia and obesity in this population more accurately [50,77,78]. However, only a few studies have evaluated the impact of such measures on the clinical outcomes of adult patients with cancer [77,79,80]. Moreover, the prognostic significance of imaging-based measures in children with cancer needs to be further explored, also addressing the potential harms, i.e., those related to exposure to radiation [80]. Additionally, while in developing countries, “underweight” is certainly occurring in the normal population at a certain high percentage, “overweight” is basically the norm in highly developed countries. For example, the USA population shows an overweight prevalence of >40% [81]. So, maybe the occurrence of underweight and overweight children with cancer reflects the situation in the respective countries and is hence a “larger problem”. Regarding the outcome assessment, the identified studies assessed the impact of nutritional status on various clinical endpoints, such as OS, EFS, TRT, or specific treatment-related toxicities (cardiotoxicity, nephrotoxicity, etc.). Such heterogeneity did not allow the quantitative synthesis of the results in the context of a meta-analysis. Lastly, the retrospective design of several studies is another limitation, which increases the risk of misclassification bias; indeed, misclassification might be differential in retrospective studies, leading to recall bias, namely reports of higher exposures in children with the disease.

## 5. Conclusions

Despite the large heterogeneity of published literature and several other limitations reported herein, nutritional status seems to play a crucial role in the survival and clinical outcomes of children with cancer, thus providing a significant modifiable prognostic tool in childhood cancer management. Interventions targeting the optimal nutrition and normal development of children with cancer are thus necessary. Future studies with adequate power and longitudinal design are still needed to further evaluate the association of nutritional status with childhood cancer outcomes using a more standardized definition to measure nutritional status in this population. The use of new technologies to longitudinally and accurately assess the nutritional status on a large scale from diagnosis onwards is expected to shed further light on this understudied area and to give room to individualized person-targeted intervention strategies.

## Figures and Tables

**Figure 1 diagnostics-12-02357-f001:**
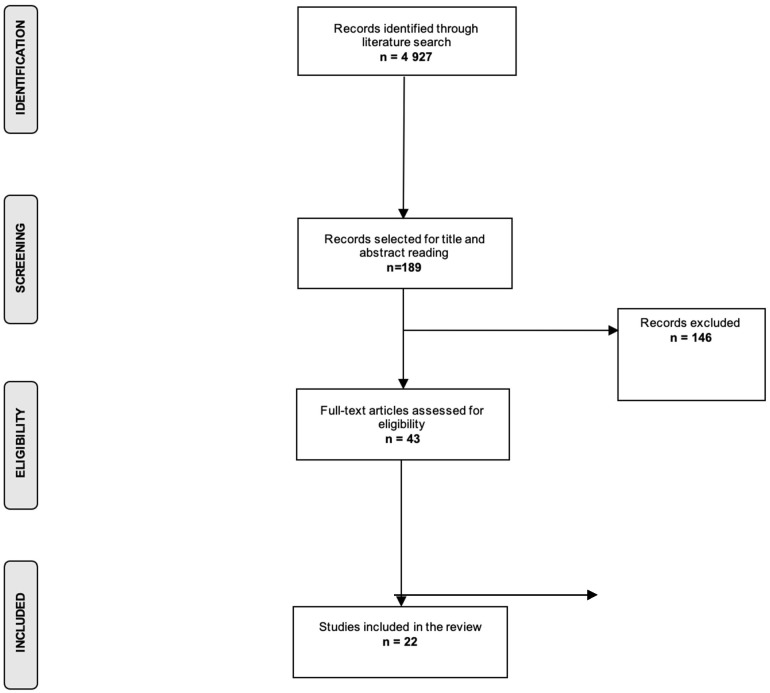
Flow chart of the selection process.

**Table 1 diagnostics-12-02357-t001:** Summary of studies reporting the association between nutritional status at diagnosis and outcomes of children with cancer *.

Cancer Site	N Studies	Outcomes
Leukemia	Meta-analyses (N = 5); 5 more recent original studies	- ALL: increased BMI associated with higher mortality, risk of relapse, and poorer EFS - AML: increased BMI associated with poorer OS and EFS - Post HSCT: malnutrition linked to poorer OS, progression-free survival, and higher risk of GvHD; lower BMI associated with poorer OS and EFS
Hodgkin lymphoma	2	- Malnutrition associated with worse OS (75–79%)
Ewing sarcoma	3	- Undernutrition associated with increased cardiotoxicity risk - No association between low BMI and TRT - Abnormal BMI (high or low) associated with poor histological response and OS
Osteosarcoma	2	- High BMI associated with increased risk of complications (arterial thrombosis, nephrotoxicity) - Low BMI associated with increased risk of wound infection and slough - High BMI linked to worse OS and EFS
Rhabdomyosarcoma	2	- Over 10% weight loss associated with increased number of hospitalization days - Patient weight (≥50 kg) associated with worse EFS
Neuroblastoma and Wilms tumors	2	- Non-significant associations

* Abbreviations: ALL, acute lymphoblastic leukemia; AML, acute myeloid leukemia; BMI, body mass index; EFS, event-free survival; GvHD, graft versus host disease; HSCT, hematopoietic stem cell transplantation; OS, overall survival; TRT, treatment-related toxicity.

## Data Availability

No additional data are available.

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
