# Peer review of "Nutritional Status at Diagnosis as Predictor of Survival from Childhood Cancer: A Review of the Literature"

_diagnostics, 2022, doi:10.3390/diagnostics12102357_

Round 1
Reviewer 1 Report
The authors describe the nutritional status at diagnosis as a predictor of survival from childhood cancer.
The nutritional status is an important, modifiable factor beside other clinical and molecular indicators in pediatric malignancies.
This review provides a useful overview to the topic, although the number of papers regarding this topic in childhood cancer is quite limited.
However, there are several points to be addressed.
Figure 1.: Some letters are upside-down, this is probably a formatting problem.
The authors confuse the readers several times with the reporting of non-statistically significant findings (for example, references 18, 19, 22, 24, 28, 29, 30, 32, 33).
This makes this review somehow arbitrary.
The authors should skip all the variables with non-significant outcomes, or indicate clearly that this factor had no impact.
Additionally, in several paragraphs the structure is confusing.
Ewing sarcoma: The authors describe in the first sentence the effects of BMI on TRT, then on the the histologic response, and in the next sentence on cardiotixicity.
Similar, in the osteosarcoma sarcoma section, the outcome is followed by arterial thrombosis.
It would be much clearer if there would be transitional words in between the different parameters.
Reviewer 2 Report
The authors reviewed the literature on a rarely published theme which is if and what impact does the nutritional status of children with cancer have on the outcome (survival, relapse) and treatment-related toxicity.
Specific Points of Criticism and Suggestions for Alterations:
(1) British English and American English have different spellings for „leukaemia“ (British English) and „leukemia“ (American English). In this manuscript both spellings are used mixed. While both spellings are certainly correct, only one type of spelling should be used consistently in the same paper.
(2) When percentages are used in the text, it is sufficient to use whole numbers (e.g. 75%) and it is not necessary to include digits after the komma = decimal points (e.g. 75.5%) which gives a false sense of accuracy. For example in the text on lines 116-118 and lines 222-225.
(3) In Figure 1 the grey-shaded boxes on the left side are upside down.
(4) Line 323: „hepatotoxicity“ (not hepatoxicity).
(5) Line 35: „area“.
(6) The authors might want to consider the following in their discussion: While in developping countries „underweight“ is certainly occurring in the normal population at a certain high percentage, „overweight“ is basically the norm in highly developped countries. For example, the USA population shows an overweight prevalence of >40% (https://www.cdc.gov/obesity/data/adult.html). So, maybe the occurrence of underweight and overweight children with cancer reflects the situation in the respective countries and is hence a „larger problem“.
(7) If at all possible, it would be very useful to somehow present the results in a graphical form which certainly would enhance the message of the paper.
Reviewer 3 Report
The paper is well written and highlights a very interesting issue, i.e. the opportunity to act on a variable, nutritional status, that can have an impact on final outcome.
Since the pediatric oncology community is looking at more and more issues, I would recommend to add a reference also to the long term outcome of children with cancer, eventually citing this recent report: Bagnasco F. Eur J Cancer. 2019 Mar;110:86-97. doi: 10.1016/j.ejca.2018.12.021.
Round 2
Reviewer 1 Report
Agree